# Incorporation of CD55 into the Zika Viral Envelope Contributes to Its Stability against Human Complement

**DOI:** 10.3390/v13030510

**Published:** 2021-03-19

**Authors:** Zahra Malekshahi, Sarah Bernklau, Britta Schiela, Iris Koske, Zoltan Banki, Karin Stiasny, Claire L. Harris, Reinhard Würzner, Heribert Stoiber

**Affiliations:** 1Institute of Virology, Medical University of Innsbruck, 6020 Innsbruck, Austria; Zahra.malekshahi@i-med.ac.at (Z.M.); sarah.bernklau@gmx.de (S.B.); britta.schilea@online.de (B.S.); iris.koske@i-med.ac.at (I.K.); zolta.banki@i-med.ac.at (Z.B.); 2Center for Virology, Medical University of Vienna, 1090 Vienna, Austria; karin.stiasny@meduniwien.ac.at; 3Translational & Clinical Research Institute, Newcastle University, Newcastle upon Tyne NE1 7RU, UK; Claire.Harris@newcastle.ac.uk; 4Institute of Hygiene & Medical Microbiology, Medical University of Innsbruck, 6020 Innsbruck, Austria

**Keywords:** Zika virus, complement, virolysis, CD55

## Abstract

The rapid spread of the virus in Latin America and the association of the infection with microcephaly in newborns or Guillain–Barré Syndrome in adults prompted the WHO to declare the Zika virus (ZIKV) epidemic to be an international public health emergency in 2016. As the virus was first discovered in monkeys and is spread not only by mosquitos but also from human to human, we investigated the stability to the human complement of ZIKV derived from mosquito (ZIKVInsect), monkey (ZIKVVero), or human cells (ZIKVA549 and ZIKVFibro), respectively. At a low serum concentration (10%), which refers to complement concentrations found on mucosal surfaces, the virus was relatively stable at 37 °C. At higher complement levels (up to 50% serum concentration), ZIKV titers differed significantly depending on the cell line used for the propagation of the virus. While the viral titer of ZIKVInsect decreased about two orders in magnitude, when incubated with human serum, the virus derived from human cells was more resistant to complement-mediated lysis (CML). By virus-capture assay and Western blots, the complement regulator protein CD55 was identified to be incorporated into the viral envelope. Blocking of CD55 by neutralizing Abs significantly increased the sensitivity to human complement. Taken together, these data indicate that the incorporation of CD55 from human cells contributes to the stability of ZIKV against complement-mediated virolysis.

## 1. Introduction

Zika virus (ZIKV) is a mosquito-borne flavivirus, which was first isolated from monkeys in Zika forest, Uganda 1947 [1,2]. In the beginning, the impact of the virus on public health was limited with fewer than 20 human infections documented for half a century [3]. In 2007, the first outbreak in the Yap Island was reported [4], followed by the emergence of ZIKV in French Polynesia and other Pacific islands [5]. From 2015 onward, ZIKV spread to the Americas with severe consequences in pregnancy, such as microcephaly and neurological impairment of the newborns [5]. In adults, ZIKV can cause Zika fever and is associated with Guillain–Barre syndrome [6]. Although the transmission of the virus declines since 2017, ZIKV still circulates and remains a potential threat in some regions [7].

Similar to related members of the Flaviviridae family, ZIKV contains a single-stranded RNA of positive polarity with a size of about 11 kb. The RNA is directly translated by the ribosomes as a polyprotein and cleaved by viral and host-encoded proteases into seven non-structural (NS) and three structural proteins, including the capsid, membrane (prM/M), and envelope (E) protein [8]. Two flavivirus proteins are characterized as the main participants in the interaction with the immune system. NS1 functions as a regulator of viral transcription and has been shown to antagonize the antiviral immune response by interfering with the interferon pathway [9,10]. In addition, NS1 interacts with several proteins of the complement system [11,12,13]. The E protein binds to the cell surface and mediates fusion after endocytic virus uptake. The majority of the neutralizing antibody responses are directed against the E protein. Besides NS1, ZIKV-E protein also interacts with several proteins of the complement system [13,14].

As part of innate immunity, the complement system contributes to the control of pathogens from the very beginning of infection [15,16,17]. Its function is to clear invading microbes by either forming a lytic membrane-attack complex on the surface of the pathogens, or by tagging them with complement C3 cleavage products for opsonization, and attraction of immune cells. Both latter mechanisms markedly support phagocytosis by macrophages and dendritic cells and thus enhance antigen presentation to cells of the adaptive immune system. Therefore, the complement system bridges innate and adaptive immunity [15,16]. Specific proteins sense invading microbes and can trigger one of the three complement pathways. The classical pathway (CP) is activated by immune complexes, pentraxins, or by direct interaction of C1 with the surface of pathogens. Activation of the lectin pathway (LP) is mediated by MBL, ficolins, or collectins binding to glycosylated or acetylated residues on the envelope of the invading microorganism. Spontaneous hydrolysis of a labile thioester bond converts C3 to a bioactive form C3(H_2_O) and keeps the alternative pathway (AP) constitutively active at low-levels in the plasma. When activated by viruses, fungi, bacteria, parasites, cobra venom, immunoglobulin A, or polysaccharides, the generated C3bBb complex acts as a C3 convertase and generates more C3b through an amplification loop [18]. Independently of the pathway, complement activation converges at the terminal pathway, resulting in the generation of the membrane attack complex (MAC) potentially causing lysis and thus clearance of the invading microorganism [15,16,19]. To avoid lysis of the host cells, the complement system is tightly regulated. Besides regulators of complement activation (RCAs) in fluid phase, membrane-anchored RCAs such as CD46, CD55, and CD59 interfere at different levels of the complement cascade [20]. CD55 for example destabilizes the C3- and C5-convertases. Consequently, the terminal complement pathway and thus the formation of the MAC are blocked.

To avoid destruction by the complement system, viruses have acquired several escape mechanisms, such as inactivation by enzymatic degradation, recruitment or mimicking of complement regulators, or inhibition of complement proteins by direct interactions [21,22], as shown for retroviruses, orthopox, or herpes viruses to name only a few [23,24,25,26,27]. In addition, Flaviviridae has adapted strategies to escape lysis by the complement system [11,12,28]. By NS1 and the E protein, ZIKV interferes with the formation of the MAC, which contributes to the resistance against the lower concentration of human complement proteins found on mucosal surfaces [14,29]. As the stability of viruses against complement is not only dependent on intrinsic viral factors but also on the cells the progeny virus is derived from [30], we tested whether ZIKVs propagated in different cell lines differed in their stability against complement-mediated lysis. ZIKV is spread by mosquitos and can be transmitted to monkeys and humans and also from human to human [2,3]. Therefore, an insect cell line (C6/36), monkey cells (Vero), and human cells (A549 and primary fibroblasts) were used to produce progeny virus, which gave rise to ZIKV_Insect,_ ZIKV_Vero_, ZIKV_A549,_ and ZIKV_Fibro_, respectively.

The aim of the study was to assess whether human cell-derived ZIKVs are more stable to destruction and, if positive, to what extent the incorporation of membrane-anchored complement regulator proteins (mCRPs) into the viral envelope was responsible.

## 2. Material and Methods

### 2.1. Cells and Viruses

*Aedes albopictus* C6/36 mosquito cells (ATCC, Manassas, VA 20108 USA) were grown in Dulbecco’s modified Eagle’s medium (DMEM; Invitrogen, Carlsbad, CA, USA) supplemented with 10% heat-inactivated fetal calf serum (FCS), antibiotic-antimycotic solution [10,000 units/mL of penicillin, 10,000 µg/mL of streptomycin, and 25 µg/mL Amphotericin B], l-glutamine, and nonessential amino acids (Gibco, Dublin, Ireland) at 28 °C with 5% CO_2_. Primary human dermal fibroblasts (kindly provided by the group of Prof. Zschocke, Insitute of Human Genetics in Innsbruck) were cultured at 37 °C and 5% CO_2_ in DMEM supplemented with same components as described for C6/36 cells. A549 (Manassas, VA 20108 USA) and Vero cells (Vero AC-free catalog number: 08011101) (ECACC) were maintained at 37 °C in a 5% CO_2_ environment using DMEM containing 10% FBS and l-glutamine. The hybridoma line 4G2 (ATCC) was grown in Hybri-Care medium (ATCC: 46-X) supplemented with 10% FBS. ZIKV strain MRS_OPY_Martinique_PaRi_2015 (GenBank: KU647676) was provided by European Virus Archive (Marseille, France). For propagation of the ZIKV strains, cells were seeded in culture plates and infected at a confluence of about 80%. After washing with phosphate-buffered saline (PBS), ZIKV was added with a multiplicity of infection (MOI) of 0.1 and incubated 1 h at 37 °C. Then, cells were washed twice with PBS and fresh medium was added. Depending on the growth kinetics of the cell line, the supernatants were harvested and filtered through a 0.45-μm filter to remove cell debris. To generate higher titers of viral stocks, the supernatants were centrifuged overnight (Rotanta 460R, Hettich, Tuttingen, Germany; 4600 rpm, 16 h, 4 °C). Obtained supernatants were aliquoted and stored at −80 °C. For virus capture assay and Western blot analyses, samples were layered over a 0, 10, 20, 30, and 40% iodixanol step gradient (Optiprep), prepared in a cell suspension medium containing 0.85% (*wt*/*vol*) NaCl and 10 mM Tricine-NaOH, pH 7.4. Samples were centrifuged for 16 h at 175,000× *g* in a SW32 Ti swing-out rotor at 4 °C using a Beckman L-60 centrifuge. Fractions of 0.6 mL were collected from the top (fraction 1) to the bottom (fraction 14) of the centrifuge tube. Viruses containing fractions were identified by qRT-PCR and plaque assay. Pooled (Fractions 5 to 8) were used for virus capture assay or immunoblot analysis (see below) using specific mAbs as indicated.

### 2.2. Antibody Purification

The mouse pan-flavivirus antibody 4G2 expressed by hybridoma cells was purified from cell supernatants using a HiTrap Protein G HP column from GE Healthcare (Chicago, IL, USA) according to the manufacturer’s recommendations.

### 2.3. Normal Human Serum

Normal human serum (NHS) was purchased from Dunn Labortechnik GmbH (Ansbach, Germany) and stored in aliquots at −80 °C. For experimental procedures, serum was thawed only once and kept on ice. Some aliquots from the serum pool were heat-inactivated (hiNHS; 56 °C, 30 min) and served as controls.

### 2.4. Plaque Assay

In order to count the plaque-forming units (PFU) per milliliter, it is necessary to determine the optimal dilution; therefore, the virus mixture [total volume 100 μL/sample] was serially titrated using 10-fold dilutions and added to Vero cells grown in 6-well (addition of 800 μL) or 12-well (addition of 400 μL) plates. ZIKV samples were incubated with the cells for 1 h at 37 °C and layered with plaque agarose. Four days after incubation at 37 °C, the viral plaques were visualized using crystal violet staining. The viral titers were expressed as PFU/mL, calculated as [(number of plaques per well) × (dilution)]/(inoculum volume).

### 2.5. Serum-Sensitivity Assay

ZIKV [1 × 10^6^ PFU/mL] derived from different cell lines was incubated with 10%, 20%, or 50% (final concentration) NHS, heat-inactivated NHS (hiNHS), or DMEM (supplemented with FCS) as a control. When indicated, lysis assays were performed in the presence of anti-IgM Abs (Bethyl Laboratories, A80-100A, Montgomery, AL, USA) as recommended by the manufacturer to block putative interaction of natural IgM with the virus. After an incubation time of 1 h at 37 °C, all samples were serially diluted and titrated on Vero cells as described above for the Plaque Assay.

### 2.6. RT-PCR

Using the NucliSENS easyMAG (BioMérieux, Amersfoort, The Netherlands), the viral RNA extraction was performed as recommended by the manufacturer. To measure the ZIKV-specific RNA, a reverse transcription-PCR (RT-PCR) was carried out, using the iScript One-Step RT-PCR Kit (Quanta; Thermofisher, Vienna, Austria). Primer and probe sequences as well as the thermal profile of the cycler were performed in accordance with the experimental procedures reported by Lanciotti et al. [31].

### 2.7. Virus Capture Assay

Microtiter wells (Maxisorp, Nunc; Sigma-Aldrich, Vienna, Austria) were coated overnight at 4 °C with antibodies against human CD46, CD55 (BD Pharmingen, Vienna Austria) or CD59 (MEM43, generous gift of P. Morgan, Cardiff, UK; 10 µg/mL each). As positive capture control, the antibody against flavivirus envelope 4G2 was used. Wells were blocked with 5% skim milk in PBS for 1 h at 37 °C. ZIKV propagated in different cell lines (1 × 10^8^ RNA copies/mL in PBS) was added and incubated for 2 h at 37 °C. Putative contaminating viral RNA or RNA from spontaneously lysed viral particles in the supernatant was digested by the addition of 1 mg/mL RNase A (Macherey-Nagel 740505, Dueren, Germany). Wells were washed six times with PBS. To determine the relative amount of the captured virions, lysis buffer was added and RNA extracted using the NucliSENS easyMAG system (BioMérieux, Vienna, Austria) according to the recommendation of the manufacturer. Relative RNA copy number was determined by qRT-PCR. The ct-value obtained by the positive control (mAb 4G2 against ZIKV E protein) was set as “1.0” and the relative change of the ct-units is given as “fold increase”.

### 2.8. Flow Cytometric Analysis

For the detection of membrane-bound complement regulatory proteins (mCRPs), virus-producing cells were immune-stained as follow: Vero cells, primary Fibroblasts, and A549 cells (5 × 10^5^ cells/tube) were harvested by detaching with Accutase (Sigma, Vienna Austria), washed twice with buffer containing 2% FBS, 0.2% EDTA in PBS, and incubated with 10 µg/mL anti-CD59, anti-CD55, anti-CD46 or isotope control Abs for 1 h at 4 °C. After washing once in 1 mL FACS-buffer, a secondary goat anti-rabbit IgG (H + L) labeled with Pacific Blue (ORIGIN) was added and incubated for 30 min at 4 °C. Samples were washed again, fixed, and analyzed by flow cytometry on a FACS Canto II (Becton Dickinson, Vienna Austria). Expression of mCRPs was quantified by the determination of mean fluorescence intensity (MFI) counting of 1 × 10^4^ events/sample.

### 2.9. Western Blot

As serum proteins are highly concentrated, NHS-containing samples were diluted at least 1:50 with PBS, before 3X Laemmli buffer was added. Samples were boiled at 95 °C for 5 min. Equal amounts of protein (25 µg) were separated by electrophoresis on 10% SDS-PAGE and then transferred to nitrocellulose membranes (Miltenyi Biotec, Auburn, AL, USA). The nonspecific antibody-binding sites were blocked with 5% skim milk powder (SMP) in water, before primary antibodies (see list above) were added and incubated at 4 °C overnight. Subsequently, the membranes were washed three times with PBS-Tween 20 (0.05%). Blots were treated with horseradish peroxidase-conjugated secondary antibody (1:10,000; Jackson Immuno Research, West Grove, PA, USA), and signals were detected by enhanced chemiluminescence (ECL, made in-house).

### 2.10. Statistical Analyses

Statistical analyses were performed using the GraphPad Prism 8.0 software. All experiments were repeated at least three times and always performed in duplicate. The data have been tested for normality in order to orient the statistical tests. The difference between the two groups was assessed by t-test and Kruskal–Wallis correction. When comparing more than two groups, ANOVA followed by Bonferroni post-hoc tests or Sidak’s multiple comparisons was performed. A 95% significance level (*p* < 0.05) was considered statistically significant (* < 0.05, ** < 0.01, *** < 0.001, and **** < 0.001).

## 3. Results

### 3.1. Cell Line-Dependent ZIKV Lysis

As ZIKV can be transmitted not only by mosquitos but also by human–human interactions, we were interested in whether the complement-mediated lysis of virions would be affected by cell-line-associated factors. For this purpose, we investigated the serum stability of ZIKV stocks derived from A549 cells (ZIKV_A549_), Vero cells (ZIKV_Vero_), or primary fibroblasts (ZIKV_Fibro_) and compared them with ZIKV from insect cells (ZIKV_Insect_). The addition of 10% NHS delivered comparable data without any significant reduction in virion number, independent of the replication of ZIKV in human, monkey, or insect cells, respectively. Only a slight insignificant decrease of plaque titers was noticed when ZIKVs were incubated in 20% NHS (Figure 1A). The most dramatic difference was observed when different ZIKVs were incubated in 50% NHS. Here, ZIKV_A549_ and ZIKV_Fibro_ remained relatively stable and showed a reduction in plaque titers of only about 0.5 orders of magnitude, while viral particles produced in insect cells were neutralized by about two orders of magnitude corresponding to a 99% reduction (Figure 1A). To compare the different viral preparations among each other, we normalized on the input virus for each group individually and compared the stability in 50% NHS (Figure 1B). ZIKV derived from human cells were again less effectively cleared in NHS, while ZIKV_insect_ was effectively neutralized in this setting.

### 3.2. Complement Activation Is Not Driven by IgM

Recently, we have shown that natural IgM antibodies, most likely against insect-like structures, activate the classical pathway of complement when ZIKV is derived from insect cells [13]. Therefore, we tested whether slight titer reduction of ZIKV_A549_ and ZIKV_Fibro_ observed in human serum relates to an activation of the complement system by IgMs and are subsequently lysis by this route. While blocking of IgM resulted in a total rescue of insect-derived virus [13], ZIKV derived from human cells remained sensitive to complement exposure (Figure 2). Thus, we concluded that complement-mediated titer reduction of ZIKV derived from human cells is not due to IgM-mediated activation of the classical pathway. As a positive control, the sensitivity of ZIKV_insect_ is shown and its rescue in the presence of anti-IgM.

### 3.3. Expression of Complement-Regulators on the Cells

FACS analyses for CD46, CD55, and CD59 cells were performed to assess which mCRP is a potential candidate for uptake during viral assembly. As expected, all three mCRPs were clearly detected on infected human cells (Figure 3). Fibroblasts showed a slightly reduced expression of CD46 and CD55 when compared to A549 cells (Figure 3, dark grey histograms). In contrast, CD59 was recognized by the detection Ab equally well. Analysis of ZIKV-infected Vero cells revealed that only CD55 levels were increased on infected cells (Figure 3). As expected, the insect cells line C6/36 gave no signal with the anti-mCRP Abs used for the analysis (data not shown). No difference in the expression pattern of the mCRPs between infected and non-infected cells was observed (data not shown).

### 3.4. Incorporation of Complement-Regulator CD55 into the Viral Envelope

To analyze the putative acquisition of cellular complement regulators by ZIKV, we aimed to capture virions (ZIKV_A549_, ZIKV_Vero_, ZIKV_Fibro,_ or ZIKV_Insect_) by immobilized antibodies against human-CD46, CD55, CD59, or isotype control. As a positive control, antibodies against ZIKV envelope protein were used. While particles with incorporated mCRPs should bind to their specific antibody, viral particles, in which mCRPs are absent, should not be captured and are removed by washing steps. After viral lysis of captured virus, the extracted RNA was analyzed to determine the threshold-cycles by qRT-PCR. Despite a non-specific background, seen for mCRP-capture with ZIKV_Insect_, all mammalian-derived ZIKV particles were clearly captured using anti-human-CD55 antibody, with the strongest signal when ZIKV_A549_ was used (Figure 4). Neither Abs against CD46 nor against CD59 were able to capture ZIKV independent of the cell line that the virus was derived from. As the strongest signal was obtained using anti-CD55 with virus harvested from A549 cells, Western blot analyses were performed with this isolate and compared with ZIKV_Vero_ or ZIKV_Insect_. The association of CD55 with A549-derived particles could be further confirmed by Western blot analysis. As expected, ZIKV derived from cells of non-human origin showed no signal in the blot (Figure 5). As a control, an antibody against the ZIKV E protein was used (Figure 5). In contrast to CD55, we could not detect CD59 by Western blot independent from the virus used for the analysis (data not shown).

### 3.5. Membrane-Bound Regulators Provide Additionally Serum Resistance

The results demonstrated that CD55 was incorporated in the viral envelope and we hypothesized that this virus-associated regulator contributed, at least partially, to serum resistance. Thus, we hypothesized that the treatment of ZIKV_A549_ with a CD55-blocking antibody might enhance the sensitivity to human complement, and decrease resistance of the virus to human serum. To test this, ZIKV_A549_ was incubated either with anti-human CD55 blocking antibody (HD1A) [32] or anti-human CD59 blocking antibody (MEM43) for 30 min on ice before active serum was added. As a control, an isotype control antibody was used. The analysis of ZIKV infectivity by plaque assay did not show any significant changes in infection titers between ZIKV samples treated with blocking or isotype control antibodies (results not shown). Thus, we investigated the blocking effect on RNA level using qRT-PCR, as a more sensitive approach. This showed that ZIKV RNA copies were reduced by about 25% when CD55 was inhibited (Figure 6), whereas the isotype control (Figure 6) or anti-CD59 Ab (data not shown) did not show any significant changes on RNA level.

## 4. Discussion

Our results show that the cellular origin of ZIKV particles affects their stability. Virus produced in human cells A549 or primary fibroblasts seems to be less susceptible to complement-mediated lysis when we compared ZIKV_Insect_ or ZIKV_Vero_. As reported for VSV-GP, differences in glycosylation patterns of insect-derived ZIKV do not contribute significantly to a direct enhancement of complement activation [30]. Rather, different glycoproteins allow the binding of natural antibodies directed against such xenoantigenic structures. The activation of the classical pathway could be blocked by anti-IgM [13]. As such xenoantigens are not present on A549- or fibroblast-derived virus, blocking of IgM had no effect. Therefore, we focused on components expressed on the surface of the propagating cells. To identify putative candidates for incorporation into the viral envelope, we tested for the presence of membrane-anchored complement regulators CD46, CD55, and CD59 on infected cells by flow cytometry. Interestingly, no correlation between serum stability of the virus and expression of membrane-bound regulators on the propagating cells was observed. Although CD59 was the most highly and equally expressed mCRP on both A549 cells and fibroblasts, this RCA was not detected on the viral surface. In contrast, CD55, which was also expressed on human cells, was selectively acquired by ZIKV_A549_ and ZIKV_Fibro_. This was unexpected as both the assembly of the virus and the modification of the GPI-anchored proteins CD55 and CD59 occur in the same cellular compartment, the lumen of the ER. Therefore, uptake of both complement regulators by ZIKV should be possible from a spatial point of view [33]. Although the selection mechanism is unclear, HCV, another member of the Family *Flaviviridae* and Genus *Hepacivirus,* also shows a preference for a distinct GPI-anchored protein. Here, selective incorporation of CD59 was reported, while neither CD55 nor CD46 was detected in patient- and cell culture-derived HCV isolates [28]. These findings have been corroborated by Amet and co-workers, who confirmed the CD59 association with the envelope of HCV [34]. In contrast, Mazumdar and colleagues reported not only upregulation of CD55 expression at the mRNA and protein levels shortly after an HCV infection but also an enhanced uptake of CD55 during viral assembly [35]. It should be considered that the contrasting data mentioned above might be due to differences in the reactivity of antibodies, cells lines used for virus production, or purification strategies. Therefore, we do not exclude that in other settings, CD59 may be incorporated by ZIKV. The weak interspecies cross-reactivity of mAbs for CD55 on Vero cells in the FACS analysis and the complete lack of signals for CD46 or CD59 was in line with the results obtained by the Virus capture assay. None of the antibodies against RCAs was able to retain ZIKV_Vero_ or ZIKV_Insect_.

Similar to our FACS analyses, Johnson and colleagues reported CD46 expression on A549 cells, which acts as a cofactor for cleavage and inactivation of C3b into iC3b. A similar C3b cofactor activity has been found by in vitro cleavage assays when A549-derived Paramyxoviruses Simian Virus 5 virions (SV5) were treated with NHS. While SV5 particles from CD46-expressing cells were associated with iC3b, no enzymatic cleavage event was detected when virions were generated from CD46-negative CHO cells. Additionally to SV5, the acquisition of CD46 has been demonstrated for Mumps Virus (~100–600 nm in size), leading to slower kinetics and more resistance to serum neutralization [36]. The integration of the large transmembrane domain of CD46 is probably hampered by the small size and limited surface area of Flavivirus membranes. The hypothesis of Johnson et al. that the acquisition of CD46 is limited to virions of “large membrane surface” is challenged by investigations that studied the incorporation of mCRPs by human immunodeficiency virus type 1 (HIV 1). It was already shown more than two decades ago that HIV-1 integrates not only CD55 and CD59 but also acquires CD46 at levels that protect from complement-mediated destruction [37,38,39]. Why relatively small retroviruses incorporate CD46 while ZIKV and HCV exclude this RCA remains to be determined.

Due to the pattern observed in the Western blot, a polymorphic expression of CD55 is likely. A similar expression of CD55 is reported in human colorectal cancer, which is likely to reflect variability in the o-glycosylation of the protein in tumor cells [40]. Our capture results and Western blot analyses showed that GPI-anchored proteins can be taken up by ZIKV, we were interested in whether the incorporated levels of CD55 are sufficient to provide at least partial protection against complement-mediated virolysis. For this purpose, ZIKV_A549_ was incubated with blocking antibodies against human-CD55 before serum was added. Indeed, a slight but significant reduction of the viral load (about 25% at RNA level) was recorded.

Our ZIKV capture data indicate that only CD55 is physically associated with the virus membrane, as viral particles released from the surface of infected cells could be captured by anti-human CD55 antibody, while neither CD46 nor CD59 could be detected. Taken together, we could show that ZIKV can incorporate membrane-bound regulators to enhance its stability when exposed to damaging complement components. CD55 expression at the cellular level (based on FACS data) together with incorporation of these GPI-anchored proteins into the viral envelope might explain why ZIKV_A549_ is more stable in serum than ZIKV_Insect_.

## Figures and Tables

**Figure 1 viruses-13-00510-f001:**
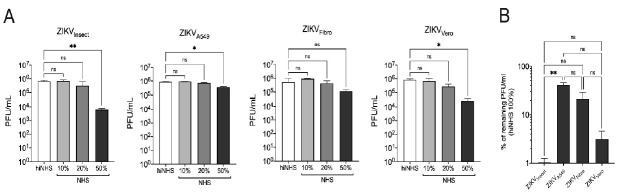
Serum stability of Zika virus (ZIKV) is affected by cell-line-dependent features. (**A**) ZIKV_Insect_, ZIKV_A549_, ZIKV_Fibro,_ and ZIKV_Vero_ were incubated with increasing amounts of active or heat-inactivated human serum. (**B**) For analysis of the different viral preparations, the input virus was normalized on the amount of virus in heat-inactivated normal human serum (hiNHS). The graph shows the stability of ZIKV derived from different cells in 50%NHS. In both sets of experiments, virus-serum mixtures were incubated for 1 h at 37 °C, then diluted 10-fold, titrated on 12-well plates of overnight-plated Vero cells and incubated for 1 h before plaque agarose was overlaid. Plaques were visualized four days post infection using crystal violet staining. All assays were performed in triplicate, and the error bars show standard deviations. A 95% significance level (*p* < 0.05) was considered statistically significant (* < 0.05, ** < 0.01).

**Figure 2 viruses-13-00510-f002:**
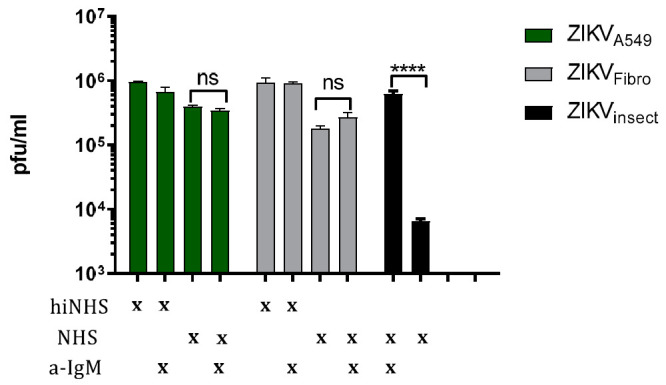
Effect of IgM on complement activation. To investigate whether natural serum IgMs affect the complement activation of human cell-derived ZIKV, anti-human IgM blocking antibodies were incubated in 50% NHS or hiNHS for 30 min on ice, before ZIKV_A549_ or ZIKV_Fibro_ were added. The experimental conditions are indicated by “x”. As positive control, ZIKV_insect_ was included [13]. After incubation for 1 h at 37 °C, the virus–serum mixture was serial diluted and titrated on 12-well plates of overnight-plated Vero cells. One hour after incubation at 37 °C, plaque agarose was overlaid. Viral concentration was determined and calculated four days post-infection using crystal violet staining. All virus lysis experiments were conducted in triplicate, and the error bars show standard deviations. A 95% significance level (*p* < 0.05) was considered statistically significant (**** < 0.001).

**Figure 3 viruses-13-00510-f003:**
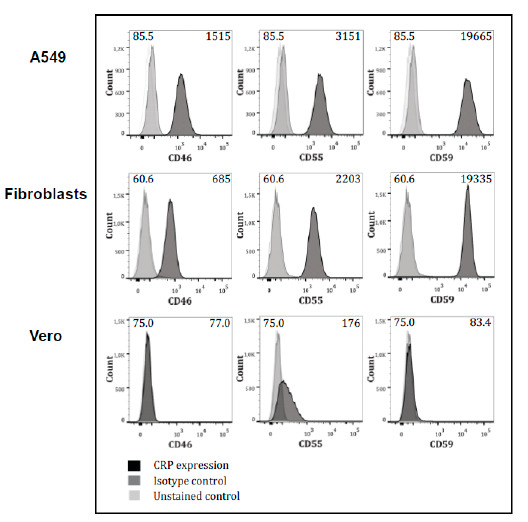
FACS analyses on A549, fibroblasts, and Vero cells for the expression of CD55, CD59, and CD46. Detection of membrane-anchored complement regulator protein (mCRP) expression on ZIKV-producing cells by FACS analysis. Cells were stained with anti-human CD46, CD55, or CD59 (dark grey histogram) or isotype-matched Ab (light gray histogram).

**Figure 4 viruses-13-00510-f004:**
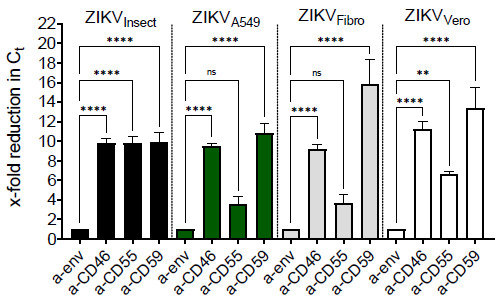
Capture ELISA indicates CD55 incorporation in ZIKV envelope. Graphs depict fold changes of RNA genome copies of captured ZIKV, analyzed by RT-PCR. Intact virions ZIKV_A549_, ZIKV_Insect_, ZIKV_Fibro_, or ZIKV_Vero_ were incubated overnight on pre-coated ELISA plates, bearing antibodies against human CD46, CD55, and CD59. ZIKV envelope served as a positive control. Isotype control was used as a negative control. After extensive washing steps, in order to remove unbound virions, ZIKV particles were lysed and RNA was extracted. The captured copy number of ZIKV genomes was characterized by determining the threshold-cycles by qPCR. Experiments were done two times in duplicates, and the error bars show standard deviations. Data were analyzed by ANOVA followed by Sidak’s multiple comparisons. A 95% significance level (*p* < 0.05) was considered statistically significant (** < 0.01, **** < 0.001).

**Figure 5 viruses-13-00510-f005:**
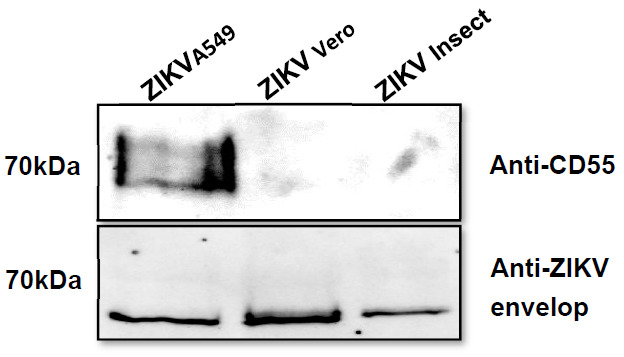
Western blot of ZIKV lysates confirms CD55 uptake. ZIKV-containing supernatants (concentrated) of infected A549, Vero, or Insect cells were characterized for CD55 incorporation by Western blotting. Samples were incubated with anti-human CD55 antibody. As load control, anti-ZIKV envelope was used. A representative Western blot analysis is shown.

**Figure 6 viruses-13-00510-f006:**
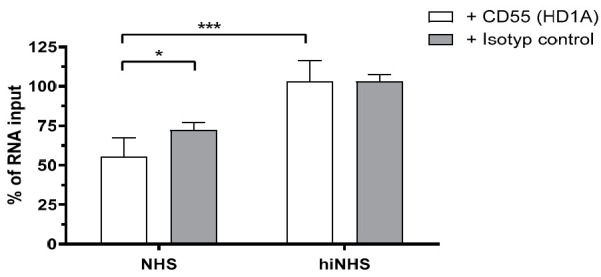
Blocking of CD55 enhances the serum sensibility of ZIKV_A549_. ZIKV_A549_ was pre-incubated with anti-CD55 hybridoma (HD1A) or isotope control antibody before active or heat-inactivated human serum was added. The viral RNA of lysed particles was destroyed by 3 h incubation with RNase A. RNA of complement-mediated lysis (CML)-intact/CML-resistant virions was extracted and analyzed by qRT-PCR. The average value of lysis data using the isotype control was set to 100%. The measurement values were calculated in relation to the average value = 100%. Data were analyzed by two-tailed *t*-test (* *p* < 0.05, *** *p* < 0.001). The assay was performed in triplicate.

## Data Availability

The data presented in this study are openly available in this published paper.

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
