# Peer review of "Incorporation of CD55 into the Zika Viral Envelope Contributes to Its Stability against Human Complement"

_viruses, 2021, doi:10.3390/v13030510_

Round 1

Reviewer 1 Report

The manuscript reports CD55 (DAF) association to the ZIKV envelope playing a role in the complement resistance. The manuscript contributes to the study of viral escape mechanisms of the destruction mediated by the complement system.

I have the following critical comments and suggestions to be considered by the authors:

(1) Page 2, lines71-74. define in this section abbreviation or acronym mCRP, which was used for the first time on line 233, page 5;

(2) It is essential to inform which kind of Vero cell from ATCC was employed in the infection assays (E6? CCl-81?);

(3) Page 5, Figure 1. Is there no statistical difference between the viral yield of ZIKVinsect and ZIKVVero in the presence of 50% NHS? Mention it in the text.

(4) Page 5 and 6. Section "Complement activation is not driven by IgM". Here the authors incubate ZIKV derived from two mammal cells (A549 and primary culture of human fibroblasts) in the presence of an anti-IgM antibody. The authors always observed a slight decrease in viral yield in the 50% NHS condition and would like to test if it would be related to the natural IgM antibodies. The effect of natural IgM antibodies was previously reported with ZIKV derived from insect cells. The data presented in Figure 2 led the authors to conclude that the effect was not due to the IgM-activated mediation. However, in my opinion, the impact of IgM incubation in insect cells infected with ZIKV should also be shown in this figure since it is positive control of the effect.

(5) Figure 5. At the end of the figure caption, the authors can mention the number of experiments made in the analyses and the meaning of the error bars;

(6) page 9, line 316,317- HCV belongs to the Family Flaviviridae (letter in italic) Genus Hepacivirus (letter in italic);

(7) ZIKV envelope protein is glycosylated. However, insect cells and mammal cells 

Both envelope proteins (E protein and prM/M protein) are N-glycosylated. It is well known that the glycosylation in insects is far simpler (high mannose and paucimannosidic N-linked structures) compared to glycosylation of higher eukaryotes (High-mannose, complex structures, and hybrids of high-mannose and complex structures). How would be the effect in complement activation based on these differences in the kind of N-glycosylation pattern? Would it be possible to explain part of the results based on these differences in the virion particle? I think it would be interesting to include this matter in the discussion section.

Reviewer 2 Report

The present manuscript investigates the role of CD55 in Zika propagation and stability. The authors found that CD55 is incorporated into the ZIKV envelope and contributes to the stability of ZIKV against complement-mediated virolysis.

I think the author’s idea is very interesting, and the way they have used mosquito, monkey, and human cells-derived virus to explore the main question was clever. I think this article deserves to be published since it brings an important finding regarding the immunological responses of humans to flaviviruses and will be of interest to the readers in the field. However, minor changes must be addressed before the publication. 

Abstract 

The abstract is well-written. It has sufficient background and clearly presents the main findings of the article.  

Introduction 

The introduction is also well-written. It has a good balance of background information and a good pace and depth for any reader to follow the work.  

Material and Methods 

Line 95: Why did the authors choose C6/36 (Aedes albopictus) instead of other Aedes cell line? The fact that they do not have RNAi pathway could have interfered in the overall outcome of the study (weakened viable particles)? Perhaps, the authors should include a commentary on this issue.  

Line 158: Is this conventional RT-PCR? I think it was done by qRT-PCR, right? Please, correct the method.  

Line 183: Have the authors verified whether each data was parametric or not? From what is described (t-test and ANOVA), I have inferred that everything was parametric. Either way, the authors should state that the data have been tested for normality in order to orient the statistical tests.  

Results 

Line 204: The significance bars are a little odd to me. The authors should highlight significance or not in the groups (hiNHS/10%/20%/50%) as was done in the final one (50%). There is no significance bar for the first group and the second is confusing (comparison between groups – which is no significant and it is already written in text (lines 199-200). For the 20%, only 2 bars have a comparison? Why are the error bars not showing the lower ranges? I would recommend improvements in this figure. 

Line 221: Figure 2 needs to be improved. It may be not clear for all readers that the “X” is presence. Please, explain the symbol in the legend. The hiNHS groups do not have a significance bar (even though they seem ns as well). ZIKVa549 is bold in the legend (Is there a meaning?). Error bar low ranges? 

Figure 3 resolution is not great, at least in my version. It can have a better resolution. 

253: Please, replace qPCR with qRT-PCR. 

264: There is a typo (“fn”) 

Line 265: The figure can be improved. Same issues with the error bars as discussed before plus the thickness that is not matching the previous figures. The asterisks could be better centered on the bars. I enjoyed the results. Very interesting assay and very convincing evidence to me. The a-env could have error bars too, correct? I know it is 1 by definition but how do the results vary among replicates? 

Line 274: Figure 5 needs to be improved. The resolution is poor and it is over-saturated (not natural) in my version. The molecular mass lane has a white spot at the bottom figure that resembles a white band/an erase tool (not affirming it is but it caused this impression). Can the authors just trim this part since the 70kDa is already indicated? The ZIKVa549 lane at the top has a weird bands’ pattern. It looks like the gel is turned 90degrees (and you see 3 bands). Is this correct? Just checking it. This weird band pattern won’t compromise the finding… Could it inform any speculation about different CD55 complexes (mass wise)? Despite the figure quality, it is an interesting and convincing result.  

Line 288: What is the potential explanation for the limited sensitivity to detect this 25% reduction/any reduction with the plaque assay? It is expected to have more RNA than viable particles, but the relationship/correlation should be preserved. The authors should discuss this result further.  How were the titration levels in comparison to the RNA levels? 2 orders of magnitude (similar to what the authors have with the titers from pfu/copies (106to 108)? 

Line 290: Please, replace RT-PCR with qRT-PCR 

Reviewer 3 Report

This manuscript describes apparent incorporation of the CD55 molecule into Zika virus virions and its apparent effect upon complement-mediated virolysis. While the observations are interesting, and there clearly are differential effects between the various CD molecules tested, the assumption that CD55 is incorporated into virions needs strengthening, and some of the methods require better description.

Major concern:

Concerning conclusion that CD55 is directly incorporated into virions, the presented data are suggestive. The plate capture using different antibodies is helpful, but the possibility that another factor is physically present in the virion envelope and responsible and that CD55 acts as a bridge has not been ruled out. This would require analysis of highly purified virions, and the methods are further confused by the described purification. How can centrifugal speeds of only 4600rpm (line 114; which should be converted to RCF) "purify" a particle as small as a flavivirus?

Minor concerns:

There are numerous grammatical errors which should be cleaned up and several figures have miss-spelled labels.

Lines 127-128; do the volumes indicated represent the cell seeding volumes or the virus inocula volumes?

Line 200 and elsewhere; presumably the percentage values for hiNHS and NHS represent final serum concentrations? This would be better indicated by replacing "with 20% NHS" with "in 20% NHS" (line 200) and elsewhere.

Line 284; hypotheses are tested, not proven.

Round 2

Reviewer 3 Report

Authors have attempted to respond to my original concerns about whether Zika virions were sufficiently purified. Their description is much improved, although the "rpm" at line 114 still needs to be converted to "rcf".

Unfortunately, the purification description (lines 114-120) does not result in virions sufficiently pure for any conclusive analysis of whether any given protein is truly associated with the particle, and this study must be rejected until the authors convincingly purify their particles. Hence, the methods used do not support the Title. As a minimum, virions should be purified by at least two orthogonal methods, not simply by a single ultracentrifugation step. Furthermore, top-fractionation is the absolute worst way to fractionate a gradient, as every fraction is contaminated by material at a lower density in the preceding fractions. Bottom- or side-puncture would have been greatly preferable.

Author Response

Unfortunately, the purification description (lines 114-120) does not result in virions sufficiently pure for any conclusive analysis of whether any given protein is truly associated with the particle, and this study must be rejected until the authors convincingly purify their particles. Hence, the methods used do not support the Title. As a minimum, virions should be purified by at least two orthogonal methods, not simply by a single ultracentrifugation step. Furthermore, top-fractionation is the absolute worst way to fractionate a gradient, as every fraction is contaminated by material at a lower density in the preceding fractions. Bottom- or side-puncture would have been greatly preferable.

We agree with the assumption of the reviewer that the purification of virions could have been improved by more sophisticated methods. In the following we summarize that the enrichment we have used is, however, entirely sufficient to reach the goals of the study:

  1. Virus containing fractions were analyzed by RT-PCR and additionally controlled by determination of PFU/ml to make sure that intact and infectious ZIKV was isolated.
  2. The virus capture assay convincingly proved that intact viral particles are captured, as free RNA (from putative trash RNA, or RNA associated with putative cell fragments) is digested with RNAse A BEFORE the addition of lysis-buffer to analyze the captured ZIKV by RT-PCR. The description of this step is now added to the manuscript (lines 164-167: “Putative contaminating viral RNA or RNA from spontaneously lysed viral particles in the supernatant was digested by the addition of 1 mg/mL RNase A [Macherey-Nagel 740505, Dueren, Germany])”. We would like to emphasize that cell-fragments, possibly still present in the preparations, would of course compete with ZIKV for binding to the Abs, resulting in a reduced detection of the virus by PCR and even underestimating the given data.
  1. The same argument can be raised for the assay, which shows an increased lysis of ZIKV in the presence of a CD55 Ab. Again, the Ab may bind to CD55 on the virus and putative contaminations by cellular fragments. After lysis, the RNA of destroyed/lysed ZIKV is in the supernatant was digested by RNAse A. In contrast RNA, which is packed and protected in intact virions is not affected (see lines 320-322). The amount of remaining ZIKV can be determined by RT-PCR.
  2. We would also like address the point that the same or very similar protocols were also used by others, see for example Steinmann et al. J. Virol. Vol 82: 7034-46, or Yin et al. J. Virol. Vol 90: 4232-42.

While agreeing to the reviewer’s point that purification could have been improved, we hope to have clarified that the enrichment is sufficient and the interpretation of the results is conclusive and justified by the data presented in the manuscript.